# Graph of Records: Boosting Retrieval Augmented Generation for Long-context Summarization with Graphs

## Abstract

Retrieval-augmented generation (RAG) has revitalized Large Language Models (LLMs) by injecting non-parametric factual knowledge. Compared with long-context LLMs, RAG is considered an effective summarization tool in a more concise and lightweight manner, which can interact with LLMs multiple times using diverse queries to get comprehensive responses. However, the LLM-generated historical responses, which contain potentially insightful information, are largely neglected and discarded by existing approaches, leading to suboptimal results. In this paper, we propose *graph of records* (**GoR**), which leverages historical responses generated by LLMs to enhance RAG for long-context global summarization. Inspired by the *retrieve-then-generate* paradigm of RAG, we construct a graph by creating an edge between the retrieved text chunks and the corresponding LLM-generated response. To further uncover the sophisticated correlations between them, GoR further features a *graph neural network* and an elaborately designed *BERTScore*-based objective for self-supervised model training, enabling seamless supervision signal backpropagation between reference summaries and node embeddings. We comprehensively compare GoR with 12 baselines on four long-context summarization datasets, and the results indicate that our proposed method reaches the best performance. Extensive experiments further demonstrate the effectiveness of GoR.

## 1 Introduction

Large Language Models (LLMs) have recently achieved remarkable performance in sorts of language modeling tasks (Achiam et al., 2023; AI@Meta, 2024). Among them, the long-context global summarization task is of great importance, which requires ultra-long context understanding capabilities of LLMs (Li et al., 2024a; Liu et al., 2024b). Current attempts to accomplish this task mainly include long-context LLMs (Touvron et al., 2023; GLM et al., 2024; Li* et al., 2023; Tworkowski et al., 2023) and retrieval-augmented generation (RAG) (Ram et al., 2023; Yu et al., 2023; Trivedi et al., 2022; Jiang et al., 2023b; Asai et al., 2023). In comparison with long-context LLMs that expand their context window to accommodate long-context inputs, RAG performs a cost-effective *retrieve-then-generate* paradigm and provides a few retrieved short text chunks from a long document to LLMs. In a running RAG system (Figure 1), there are usually a large number of historical user queries and LLM-generated responses for a long document. Nevertheless, these historical responses, which contain informative task-related content, are mostly neglected without sufficient utilization by current RAG approaches.

Unfortunately, utilizing LLM historical responses for long-context global summarization present two major challenges. (1) *Sophisticated yet implicit correlations between historical responses and text*. Given a long document, there will inevitably be complicated correlations among plentiful user queries (*e.g.*, logical correlations), which are further inherited by LLM-generated responses and the retrieved text chunks. However, uncovering these correlations is non-trivial since most text embeddings from language models (*e.g.*, SBERT (Reimers & Gurevych, 2019)) or retrievers (Karpukhin et al., 2020) concentrate on semantic similarity, which faces degrading performance in this case. (2) *Lack of supervision signal*. In contrast with local (*e.g.*, query-based) summarization (Zhong et al., 2021; Wang et al., 2022a) that includes golden reference text as labels, global summarization needs

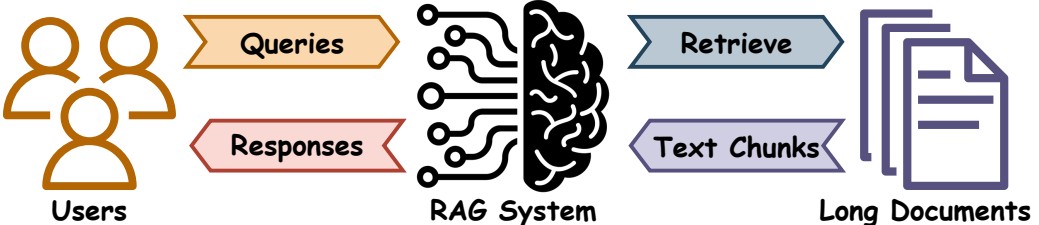

Figure 1: **Interaction between users and documents in a RAG system**. The RAG system utilizes user queries to retrieve relevant text chunks in documents, which are further fed into LLMs to generate responses back to users. In this process, massive historical responses are not sufficiently utilized.

to be considered from the perspective of the long document as a whole and only has global reference summaries, which makes it difficult to directly backpropagate effective, accurate, and deterministic supervision signals to optimize the model in the direction of a few relevant text chunks.

Based on the above observations, we propose *graph of records* (**GoR**), which utilizes and organizes LLM historical responses as a graph of records for enhancing long-context global summarization in RAG. In detail, we first leverage LLMs to simulate some user queries conditioned on arbitrary text chunks in a long document to obtain historical responses under the paradigm of RAG, and an edge is then created between the retrieved text chunks and the LLM-generated response to construct a graph of records. To learn fine-grained correlations among nodes, we employ a *graph neural network* and reuse the simulated user queries with the corresponding source text chunk as self-supervised training data. Intuitively, we hope the node embeddings can be adaptively learned to reflect the semantic and logical correlations with a given query. Inspired by the well-received BERTScore (Zhang et al., 2019) that quantifies the semantic similarity between two paragraphs of text, we rely on it to rank the nodes according to their similarity with the self-supervised label of a given simulated query. In this way, node embeddings can benefit the indirect supervision signal from the self-supervised labels and be flexibly optimized using a contrastive loss and a pair-wise ranking loss based on the node rankings. In the experiments, we adopt four long-context summarization datasets, and the results demonstrate the superiority and effectiveness of our proposed method. Our contributions are summarized as follows:

- We propose *graph of records* (**GoR**), which utilizes and organizes LLM-generated historical responses as a graph of records to strengthen RAG for long-context global summarization. We reveal that the fine-grained correlations between LLM historical responses and text chunks from long documents can be uncovered and utilized effectively to improve RAG performance.

- We leverage a *graph neural network* and design a *BERTScore*-based objective to optimize node embeddings, which can be adaptively learned in a self-supervised manner to reflect the semantic and complex correlations with input queries. Furthermore, the indirect supervision signal from self-supervised labels is crucial and conducive to the effective optimization of node embeddings.

- We evaluate our proposed method on four long-context summarization datasets, and the experimental results show that GoR outperforms several competitive baselines by a significant margin. Extensive experiments and detailed analysis verify the superiority of GoR.

## 2 GRAPH OF RECORDS

In this section, we first present some necessary backgrounds in Section 2.1. Then, we describe our proposed method sequentially through two sections, *i.e.*, Graph Construction (Section 2.2) and BERTScore-based Objective for Self-supervised Training (Section 2.3).

## 2.1 PRELIMINARIES

**Retrieval-augmented Generation**. Retrieval-augmented Generation (RAG) (Ram et al., 2023) can typically be summarized into the following two processes. (1) *Retrieval*. Give a long document which consists of several split text chunks $\mathbf{C} = \{\mathbf{c_i}\}_{i=1}^{|\mathbf{C}|}$ as retrieval corpus, RAG first employs a retriever (*e.g.*, Contriever (Izacard et al., 2021)) to retrieve $\mathbf{K}$ text chunks that are most relevant to a given query $\mathbf{q}$ based on semantic similarity. The retriever typically embeds the query $\mathbf{q}$ and a text chunk $\mathbf{c}$ from $\mathbf{C}$ using a query encoder $\mathbf{E_q}(\cdot)$ and a context encoder $\mathbf{E_c}(\cdot)$, respectively, and quantify their semantic similarity by the dot product operation described as $\mathrm{Sim}(\mathbf{q}, \mathbf{c}) = \mathbf{E_q}(\mathbf{q})^T \cdot \mathbf{E_c}(\mathbf{c})$. (2) *Generation*. The retrieved text chunks are fed into LLMs with the query $\mathbf{q}$ to obtain the final response $\mathbf{r}$. The whole process can be described as:

$$\mathbf{r} = \mathrm{Generation}(\mathbf{q}, \{\mathbf{c_1}, \cdots, \mathbf{c_K}\}), \quad \{\mathbf{c_1}, \cdots, \mathbf{c_K}\} = \mathrm{Retrieval}(\mathbf{q}|\mathbf{C}) \tag{1}$$

**Graph Neural Networks**. Graph Neural Networks (GNNs) (Kipf & Welling, 2016) stand out for their excellent representation learning ability on graph data. GNNs update node embeddings iteratively by aggregating messages from their neighboring nodes. Generally, the $l$-th layer of GNNs can be formalized as:

$$\mathbf{h}_v^{(l)} = \mathrm{AGG}^{(l)}\left(\mathbf{h}_v^{(l-1)}, \mathrm{MSG}^{(l)}\left(\{\mathbf{h}_u^{(l-1)}, u \in N(v)\}; \theta_m^l\right); \theta_a^l\right) \tag{2}$$

where $\mathbf{h}_u^{(l)} \in \mathbb{R}^{d_l}$ is the embedding vector of nodes $u$ in layer $l$ and the embedding dimension is $d_l$. $\mathrm{MSG}^{(l)}(\cdot)$ is a message computation function parameterized by $\theta_m^l$ and $\mathrm{AGG}^{(l)}(\cdot)$ is a message aggregation function parameterized by $\theta_a^l$ in layer $l$.

## 2.2 GRAPH CONSTRUCTION

In this section, we describe how to organize LLM historical responses into a graph of records by simulating user queries.

**Query Simulation.** User queries play a very critical role in the design of GoR since LLM historical responses generated by lots of repetitive, nonsense, or meaningless questions are inherently not beneficial for summarization. One solution is to use doc2query (Nogueira et al., 2019) to simulate queries for a long document, but the generated results inevitably suffer from simplicity and rigidity due to the limited text generation capabilities of T5 (Raffel et al., 2020). To this end, we directly turn to LLMs for query simulation with temperature sampling instead of greedy decoding for generating meaningful, insightful, and diverse questions. Specifically, we split a long document into several text chunks $\mathbf{C}$ following the standard procedure of RAG and prompt LLMs to generate a query $\mathbf{q^s}$ based on a randomly selected text chunk $\mathbf{c^s}$. We repeat the above process until a certain number of non-duplicate queries are generated, which are gathered in pairs with the corresponding text chunks to form a corpus $\mathbf{T} = \{(\mathbf{q_i^s}, \mathbf{c_i^s})\}_{i=1}^{|\mathbf{T}|}$ for further model training (Section 2.3). The prompt for query simulation can be found in Appendix C.

**Organize LLM Historical Responses into A Graph**. After obtaining simulated queries, we utilize them to perform RAG on the long document. LLM-generated responses during this process include informative and valuable understanding, summarizing, and answering of retrieved text chunks in the long document. Moreover, since there may exist sophisticated correlations among simulated queries, the text chunks and responses can inherit these features and potentially assist in answering a more comprehensive query, especially global summarization that needs to be understood from a holistic perspective. Nevertheless, it is a significant challenge to find correlations among complex and massive text at the linguistic level and the embeddings from language models (*e.g.*, SBERT (Reimers & Gurevych, 2019)) or retrievers (Karpukhin et al., 2020) focus on semantic similarity, which also suffers from poor performance in this case. To this end, we propose to break out of this dilemma by organizing these historical responses into a graph.

Inspired by the *retrieve-then-generate* process of RAG, we can connect the retrieved chunks to the corresponding response generated by LLMs since they are naturally relevant in content. Sequentially, during the $\mathbf{i}$-th round RAG, given the simulated query $\mathbf{q_i^s}$, we expand the retrieval corpus $\mathbf{C}$ with previously generated responses $\{\mathbf{r_1}, \cdots, \mathbf{r_{i-1}}\}$ and then build an edge between each retrieved chunk $\mathbf{c_j} \in \{\mathbf{c_1}, \cdots, \mathbf{c_K}\}$ and the newly generated LLM response $\mathbf{r_i}$, resulting in $\mathbf{K}$ edges constructed in each round. Note that we append the responses generated by each round of RAG to the

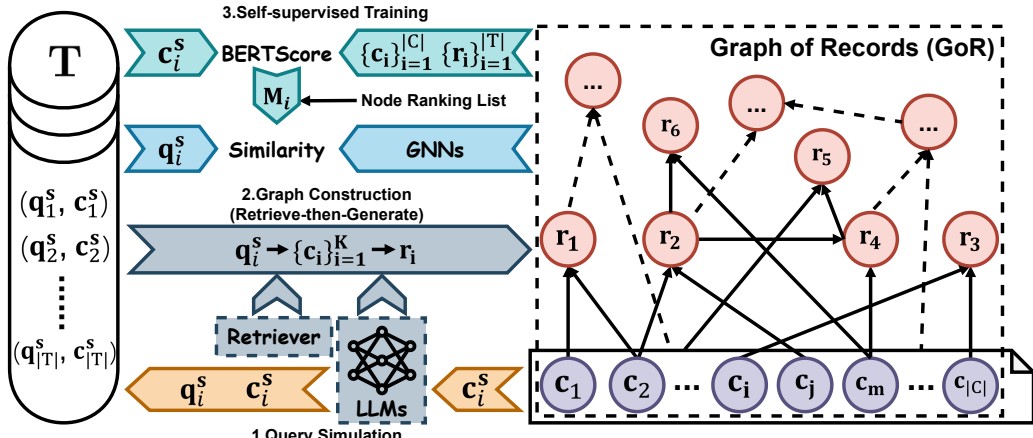

Figure 2: **GoR model architecture**. GoR randomly selects text chunks $c_i$ from long documents to feed into LLMs for query simulation, which are saved as a self-supervised training corpus $\mathbf{T}$ and further used for graph construction inspired by the retrieve-then-generate paradigm in RAG. For model training, GoR leverages GNNs to obtain node embeddings and calculate their similarities to the query embedding. Finally, GoR features contrastive learning and pair-wise ranking objectives based on the node ranking list $\mathbf{M}_i$ derived from BERTScore calculation.

retrieval corpus because they contain more refined knowledge compared with the text chunks from $\mathbf{C}$ and can help LLMs generate comprehensive responses in a self-evolving manner. Formally, the i-th round RAG on simulated queries $\{\mathbf{q}_i^s\}_{i=1}^{|\mathbf{T}|}$ can be described as:

$$\mathbf{r_i} = \text{Generation}(\mathbf{q_i^s}, \{\mathbf{c_1}, \cdots, \mathbf{c_K}\}), \quad \{\mathbf{c_1}, \cdots, \mathbf{c_K}\} = \text{Retrieval}(\mathbf{q_i^s} | \mathbf{C}, \{\mathbf{r_1}, \cdots, \mathbf{r_{i-1}}\}) \quad (3)$$

In this way, the LLM-generated responses serve as *bridges* to connect the originally scattered text chunks $\mathbf{C}$ so that the fine-grained and sophisticated correlations among them can be better modeled and explored. Furthermore, we can potentially make sufficient use of historical responses generated by LLMs and improve the quality of future LLM responses.

### 2.3 BERTSCORE-BASED OBJECTIVE FOR SELF-SUPERVISED TRAINING

So far, we have constructed a graph using LLM-generated historical responses during RAG given the simulated queries. The key in this section lies in designing a reasonable and effective objective function for model optimization. Considering that some random walk (Grover & Leskovec, 2016) or propagation-based (Zhu & Ghahramani, 2002) algorithms are not differentiable, we turn to graph neural networks (GNNs) for learning node embeddings, which are backpropagation-friendly. Intuitively, given a global summarization query $\mathbf{q}$, our ultimate optimization goal is to make the learned node embeddings adaptively reflect the similarity with the query embedding $\mathbf{E_q}(\mathbf{q})$ by taking the complicated correlations among nodes into account. However, in global summarization tasks, there are essentially no text chunk indices as labels to indicate which nodes are most relevant for a query since it needs to consider the long document as a whole. Another naive solution is to use global reference summaries as labels, but there is a gap in supervision signal backpropagation between them and node embeddings because we still need to find out which nodes are most relevant to them.

Therefore, inspired by BERTScore (Zhang et al., 2019), which measures the semantic similarity between the reference and the generated text, we propose to use it to rank all nodes based on the similarity with reference summaries. By this means, BERTScore fills the gap in the backpropagation so that node embeddings can benefit the indirect supervision signal from the reference summaries. Nevertheless, global reference summaries contain broad information about long documents, making them highly semantically relevant to many nodes, which will confuse the model optimization direction and degrade the performance (we will discuss it in Section 3.4).

**Contrastive Loss Driven by BERTScore**. Based on the above observations, we directly reuse the simulated queries $\mathbf{T} = \{(\mathbf{q}_i^{\mathbf{s}}, \mathbf{c}_i^{\mathbf{s}})\}_{i=1}^{|\mathbf{T}|}$ to serve as self-supervised training data, in which the text chunk $\mathbf{c}_i^{\mathbf{s}}$ is highly relevant to the query $\mathbf{q}_i^{\mathbf{s}}$ and has more focused content. Given node embeddings output by the last $L$-th layer of GNNs, for the $i$-th query $\mathbf{q}_i^{\mathbf{s}}$, we rank them according to the similarity with the $i$-th text chunk $\mathbf{c}_i^{\mathbf{s}}$ and obtain a node embedding ranking list $\mathbf{M}_i$, which can be described as:

$$\mathbf{M}_i = [\mathbf{h}_+^{(L)}, \mathbf{h}_1^{(L)}, \cdots, \mathbf{h}_{|\mathbf{C}|+|\mathbf{T}|}^{(L)}] \tag{4}$$

where $\mathbf{h}_+^{(L)}$ stands for the node embedding with highest similarity. Note that we utilize the context encoder $\mathbf{E}_{\mathbf{c}}(\cdot)$ from the retriever to initialize node embeddings for simplicity. Then, we regard $\mathbf{h}_+^{(L)}$ as the positive while the rest in $\mathbf{M}_i$ as negative samples to conduct contrastive learning using InfoNCE (van den Oord et al., 2018), which brings the query $\mathbf{q}_i^{\mathbf{s}}$ and the positive sample $\mathbf{h}_+^{(L)}$ closer in the semantic embedding space. We formulate contrastive training objective as follows:

$$\mathcal{L}_{\mathrm{CL}} = -\frac{1}{|\mathbf{T}|} \sum_{j=1}^{|\mathbf{T}|} \log \frac{\exp\left(\mathbf{E}_{\mathbf{q}}(\mathbf{q}_j^{\mathbf{s}})^{\top}\mathbf{h}_+^{(L)}/\tau\right)}{\exp\left(\mathbf{E}_{\mathbf{q}}(\mathbf{q}_j^{\mathbf{s}})^{\top}\mathbf{h}_+^{(L)}/\tau\right) + \sum_{i=1}^{|\mathbf{M}_j-1|} \exp\left(\mathbf{E}_{\mathbf{q}}(\mathbf{q}_j^{\mathbf{s}})^{\top}\mathbf{h}_i^{(L)}/\tau\right)} \tag{5}$$

where $\tau$ is the temperature coefficient. Note that in the optimization pipeline of GoR, we conduct mini-batch training on the graph level, and each graph is associated with an independent self-supervised training dataset $\mathbf{T}$. We also leverage in-batch negatives from other graphs since the nodes in them are completely irrelevant content from other long documents (it is not shown in Formula 5 for brevity).

**Auxiliary Pair-wise Ranking Loss**. In the above-described contrastive loss $\mathcal{L}_{\mathrm{CL}}$, although we impose constraints on positive and negative samples, the ranking of negative samples themselves is not well utilized. Inspired by LambdaRank (Burges, 2010), we further introduce an auxiliary pair-wise ranking loss on the ranking list $\mathbf{M}_i$, which can be formulated as:

$$\mathcal{L}_{\mathrm{RANK}} = \frac{1}{|\mathbf{T}|} \sum_{k=1}^{|\mathbf{T}|} \sum_{\mathbf{h}_i^{(L)}, \mathbf{h}_j^{(L)} \in \mathbf{M}_k} \mathbb{I}_{\mathrm{r}(\mathbf{h}_j^{(L)}) > \mathrm{r}(\mathbf{h}_i^{(L)})} \log\left(1 + e^{\mathbf{E}_{\mathbf{q}}(\mathbf{q}_k^{\mathbf{s}})^{\top}\mathbf{h}_j^{(L)} - \mathbf{E}_{\mathbf{q}}(\mathbf{q}_k^{\mathbf{s}})^{\top}\mathbf{h}_i^{(L)}}\right) \tag{6}$$

where $\mathrm{r}(\cdot)$ denotes the ranking index (*e.g.*, $\mathrm{r}(\mathbf{h}_+^{(L)}) < \mathrm{r}(\mathbf{h}_1^{(L)})$). Given $\mathbf{h}_i^{(L)}, \mathbf{h}_j^{(L)} \in \mathbf{M}_k$ that satisfies $\mathrm{r}(\mathbf{h}_j^{(L)}) > \mathrm{r}(\mathbf{h}_i^{(L)})$, the pair-wise ranking loss will explicitly optimize in the direction of $\mathbf{E}_{\mathbf{q}}(\mathbf{q}_k^{\mathbf{s}})^{\top}\mathbf{h}_j^{(L)} < \mathbf{E}_{\mathbf{q}}(\mathbf{q}_k^{\mathbf{s}})^{\top}\mathbf{h}_i^{(L)}$, thus imposing stricter constraints to the pair-wise ranking.

**Overall Training Objective**. To sum up, the overall training objective can be formulated as:

$$\mathcal{L} = \mathcal{L}_{\mathrm{CL}} + \alpha \cdot \mathcal{L}_{\mathrm{RANK}} \tag{7}$$

where $\alpha \in [0, 1]$ is a hyper-parameter. It is worth noting that GoR's training costs are lightweight since the only trainable module is GNNs, and no human-crafted labels are needed.

## 3 EXPERIMENTS

### 3.1 EXPERIMENTAL SETUP

**Datasets**. We evaluate our proposed method on four long-context summarization datasets, *i.e.*, AcademicEval (Feng et al., 2024), QMSum (Zhong et al., 2021), WCEP (Gholipour Ghalandari et al., 2020), and BookSum (Kryściński et al., 2021). Among them, AcademicEval collects scientific papers from arXiv for abstract writing, given the long inputs of its main body. QMSum is a query-based summarization dataset, and we **only use** "*general queries*" for evaluating global summarization. WCEP is a multi-document summarization dataset about news events, while BookSum features long-form narrative summarization. For metrics, we adopt Rouge-1 (R-1), Rouge-2 (R-2), and Rouge-L (R-L) (Lin, 2004) to assess the text alignment between the reference summaries and the predicted content generated by our proposed method. More details about adopted datasets are described in Appendix A.

**Implementation Details**. Following the standard procedure of RAG, we adopt TokenTextSplitter from LangChain[1] to split each long document into text chunks. Each chunk has a size of 256, and the chunk overlapping is 32. We generate 30 queries for each long document using Mixtral-8x7B-Instruct-v0.1 (Jiang et al., 2024), and the temperature coefficient is set to 0.5 by default in the query simulation stage. For RAG, we use Contriever (Izacard et al., 2021) for query and document embedding and retrieve 6 text chunks by default, which are fed into LLaMA-2-7b-chat (Touvron et al., 2023) with greedy decoding to generate predicted summaries.

In the training stage, we initialize the graph neural network as a two-layer graph attention network (GAT) (Veličković et al., 2017), with a 768-dim hidden dimension following the default setting of most retrievers. The batch size is 32, and the max training epoch is set to 150. We use Adam optimizer for model training and gradually decay the learning rate from 1e-3 to 0 with LambdaLR scheduler. The dropout rate and loss coefficient $\alpha$ depend on the specific dataset, and we detail them in Appendix A. We implement our proposed method using PyTorch[2] and Deep Graph Library (DGL)[3], and all the experiments are conducted on a single RTX 3080 GPU. As for LLMs, we rely on API calling from Together AI[4] to obtain responses.

**Baselines**. To have a comprehensive evaluation, we compare our proposed GoR with dozens of baselines, including (1) **Random Walk based Node Embedding** (*i.e.*, Node2Vec (Grover & Leskovec, 2016)), (2) **Sparse Retriever** (*i.e.*, BM25 (Robertson et al., 2009) and TF-IDF (Ramos et al., 2003)), (3) **Dense Retriever** (*i.e.*, Contriever (Izacard et al., 2021), DPR (Karpukhin et al., 2020), Dragon (Lin et al., 2023), and Sentence-BERT (SBERT) (Reimers & Gurevych, 2019)[5]), (4) **Hybrid Retriever** (*i.e.*, BM25+DPR with Reciprocal Rerank Fusion), (5) **Long-context LLMs** (*i.e.*, Gemma-8K (Team et al., 2024) and Mistral-8K (Jiang et al., 2023a)), (6) **Full Context** (*i.e.*, feeds all inputs to LLMs for summary generation[6]), and (7) **Thought Retriever** (Thought-R) (Feng et al., 2024).

## 3.2 MAIN RESULTS

We conduct comprehensive experiments on QMSum, AcademicEval, WCEP, and BookSum datasets compared with dozens of baselines to evaluate the long-context global summarization capabilities of our proposed method. The results are shown in Table 1.

**GoR consistently outperforms retriever-based methods**. From Table 1, our proposed GoR beats sparse retrievers, dense retrievers, and hybrid retrievers in every aspect. Thanks to the constructed graph, which integrates text chunks from long documents and LLM historical responses into a whole, node embeddings can better reflect the complicated correlations with given queries, thus significantly improving the retrieval performance of GoR. Moreover, the informative content of historical responses can also potentially enhance the summarization task.

**GoR shows superiority over long-context LLMs**. We compare Gemma-8K and Mistral-8K with a longer context window to accommodate long-context inputs. However, longer inputs may contain minor information, and long-context LLMs struggle with this situation. In contrast, GoR can effectively differentiate key and topic-related content in long texts using learned node embeddings and achieve better results with shorter input lengths.

**Additional Findings**. (1) Node2Vec produces unsatisfactory results, and the node embeddings cannot be optimized effectively since it is based on a non-differentiable algorithm. (2) Although Thought Retriever demonstrates competitive results, it is still inferior to GoR due to the lack of exploration of the correlations between retrieved text chunks and LLM-generated responses. (3) Since the context window length limit of LLMs is exceeded, "Full Context" truncates the long-context input, thus losing some information that may be important for global summarization, resulting in suboptimal results.

---

[1] https://www.langchain.com/

[2] https://pytorch.org/

[3] https://www.dgl.ai/

[4] https://www.together.ai/

[5] We use all-MiniLM-L6-v2 as the backbone.

[6] If the input length exceeds the context window limit, we randomly sample continuous text spans of maximum length multiple times to feed into LLMs and calculate the average result.

Table 1: **Experimental results on QMSum, AcademicEval, WCEP, and BookSum datasets over long-context global summarization tasks w.r.t. Rouge-L (R-L), Rouge-1 (R-1), and Rouge-2 (R-2)**. Note that the average LLM input token length of GoR and retriever-based baselines is $6 \times 256$, which is about 1.5K. (**BOLD** indicates the best score)

| Model | QMSum | | | AcademicEval | | | WCEP | | | BookSum | | |
|---|---|---|---|---|---|---|---|---|---|---|---|---|
| | R-L | R-1 | R-2 | R-L | R-1 | R-2 | R-L | R-1 | R-2 | R-L | R-1 | R-2 |
| Node2Vec | 18.5 | 31.8 | 6.3 | 19.3 | 38.3 | 10.6 | 13.9 | 20.1 | 6.3 | 13.6 | 27.4 | 4.6 |
| BM25 | 18.4 | 32.1 | 6.1 | 20.4 | 39.6 | 11.3 | 15.5 | 22.6 | 7.3 | 13.7 | 26.7 | 4.9 |
| TF-IDF | 18.3 | 31.2 | 6.3 | 19.5 | 38.0 | 10.6 | 15.3 | 22.3 | 7.3 | 13.6 | 26.6 | 4.9 |
| Contriever | 19.1 | 32.7 | 7.7 | 23.6 | 44.8 | 16.0 | 15.7 | 23.5 | 7.7 | 14.4 | 29.8 | 5.5 |
| DPR | 18.6 | 32.1 | 6.7 | 20.9 | 41.4 | 13.2 | 15.6 | 22.5 | 7.5 | 13.8 | 27.1 | 4.8 |
| Dragon | 19.2 | 33.5 | 7.7 | 23.5 | 43.8 | 15.1 | 14.6 | 21.8 | 6.8 | 13.7 | 27.2 | 4.8 |
| SBERT | 19.0 | 33.0 | 7.4 | 23.4 | 45.2 | 15.8 | 13.7 | 20.5 | 5.5 | 14.4 | 29.5 | 5.4 |
| BM25+DPR | 18.3 | 31.8 | 6.6 | 19.9 | 39.0 | 10.8 | 15.7 | 22.1 | 7.6 | 14.1 | 28.9 | 5.4 |
| Gemma-8K | **19.8** | 33.5 | 7.3 | 21.9 | 42.0 | 12.9 | 15.6 | 21.9 | 7.7 | 12.8 | 23.4 | 4.2 |
| Mistral-8K | 19.6 | 31.2 | 7.2 | 21.6 | 41.6 | 13.1 | 16.7 | 24.2 | 8.8 | 13.5 | 26.2 | 5.3 |
| Full Context | 19.4 | 33.1 | 6.8 | 21.5 | 41.1 | 12.5 | 14.4 | 21.0 | 7.1 | 14.4 | 28.9 | 5.9 |
| Thought-R | 19.0 | 33.9 | 7.6 | 22.0 | 42.6 | 13.2 | 15.2 | 22.4 | 7.4 | 14.2 | 29.5 | 5.7 |
| **GoR (Ours)** | **19.8** | **34.5** | **7.8** | **24.7** | **46.5** | **17.3** | **18.1** | **25.4** | **9.2** | **14.9** | **31.5** | **6.6** |

Table 2: **Ablation study on QMSum, AcademicEval, WCEP, and BookSum datasets w.r.t. R-L, R-1, and R-2**. (**BOLD** indicates the best score)

| Variant | QMSum | | | AcademicEval | | | WCEP | | | BookSum | | |
|---|---|---|---|---|---|---|---|---|---|---|---|---|
| | R-L | R-1 | R-2 | R-L | R-1 | R-2 | R-L | R-1 | R-2 | R-L | R-1 | R-2 |
| w/o train | 18.2 | 33.0 | 7.6 | 23.3 | 45.0 | 15.5 | 15.3 | 22.4 | 7.4 | 13.7 | 27.7 | 4.7 |
| w/o $\mathcal{L}_{\text{CL}}$ | 18.4 | 33.3 | 6.9 | 23.5 | 44.9 | 15.5 | 14.7 | 21.9 | 7.2 | 14.1 | 28.8 | 5.1 |
| w/o $\mathcal{L}_{\text{RANK}}$ | 19.6 | 33.1 | **7.8** | 23.1 | 44.4 | 15.1 | 16.6 | 24.2 | 8.2 | 14.0 | 28.0 | 4.9 |
| w/o in-b neg | **19.8** | **34.7** | **7.8** | 24.5 | 46.4 | 16.5 | 17.2 | 24.9 | 8.8 | 13.3 | 26.3 | 5.2 |
| w/ sup | 18.1 | 32.3 | 6.9 | 21.4 | 43.3 | 13.9 | 15.5 | 22.8 | 7.3 | 13.8 | 29.0 | 5.2 |
| **GoR (Ours)** | **19.8** | 34.5 | **7.8** | **24.7** | **46.5** | **17.3** | **18.1** | **25.4** | **9.2** | **14.9** | **31.5** | **6.6** |

Overall, GoR achieves the best results compared with various baselines, demonstrating the effectiveness of our proposed method.

## 3.3 ABLATION STUDY

To investigate how each component of GoR contributes to its performance, we conduct an ablation experiment, and the results are shown in Table 2.

From Table 2, we can draw several conclusions. (1) Directly using the text embeddings from the retriever without training leads to degraded performance (*i.e.*, w/o train), highlighting the effectiveness of the learned node embeddings. (2) Both the contrastive loss $\mathcal{L}_{\text{CL}}$ and pair-wise ranking loss $\mathcal{L}_{\text{RANK}}$ significantly improve performance. The pair-wise ranking loss imposes stricter ranking constraints on node embeddings, making effective use of the indirect supervision signal from the self-supervised reference summaries. (3) In-batch negatives are crucial to the performance of contrastive learning. Removing in-batch negatives (*i.e.*, w/o in-b neg) leads to a significant drop in results, especially on the WCEP and BookSum datasets. (4) Compared with self-supervised training, we utilize global reference summaries as labels to conduct supervised training (*i.e.*, w/ sup), and the results are significantly worse than the self-supervised setting. We will further discuss it in Section 3.4.

In general, GoR's reasonable module design enables it to achieve superior performance.

## 3.4 DISCUSSIONS

**Impact of GNN Architectures**. GNNs play a vital role in learning node embeddings. we explore various GNN architectures to study their impact on learning node embeddings, including GCN (Kipf & Welling, 2016), SGC (Wu et al., 2019), GIN (Xu et al., 2019), and GraphSAGE (Hamilton et al., 2017). Our findings, illustrated in Figure 3, show that GAT outperforms the other architectures. This is because GAT considers the significance of neighboring nodes when updating node embeddings, allowing the model to effectively capture essential information from the nodes. Among the other architectures, GraphSAGE performs poorly due to its unstable neighbor sampling mechanism.

Overall, GAT reaches the best results, which shows that considering the importance of neighboring nodes is effective in mining complicated correlations and is critical to improving performance.

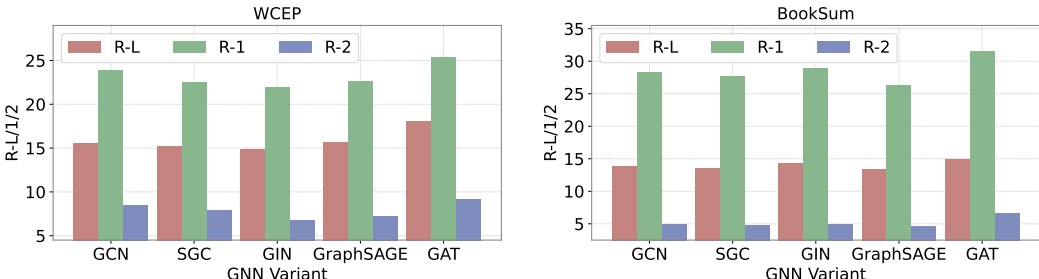

Figure 3: **Impact of GNN architectures w.r.t. R-L, R-1, and R-2**. The left figure shows results on the WCEP dataset, while the right one shows results with the BookSum dataset.

**Impact of the Number of Simulated Queries During Training**. Query Simulation is a crucial stage in our method design, and we will discuss the impact of the number of simulated queries used during training on learning performance. In particular, we explore this effect by gradually increasing the number of simulated queries used in training. We present the results in Figure 4. From a holistic perspective, R-L shows an upward trend as the number of simulated queries increases. Nevertheless, since fewer queries cover less relevant content from long documents, the curves of each dataset have some fluctuations, indicating the occurrence of underfitting.

In general, 30 simulated queries can optimize the model well across these four datasets, which indicates that our proposed GoR is cost-effective. Nevertheless, increasing the number of simulated queries may still potentially further improve the performance of the model. Due to budget constraints, we will leave this for future work.

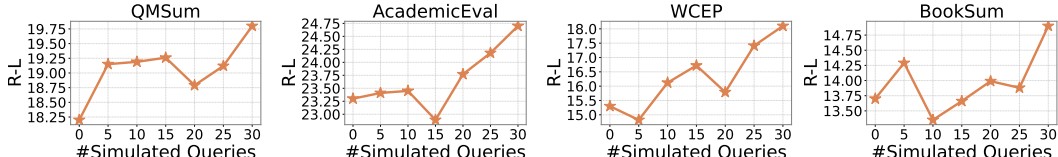

Figure 4: **Impact of the number of simulated queries (*i.e., the number of LLM-generated tokens used for training*) during training w.r.t. R-L**. We show the results on the four datasets QMSum, AcademicEval, WCEP, and BookSum from left to right.

**Supervised Training on Global Summarization Queries**. To dive deeper into the differences between self-supervised and supervised training, we carry out additional experiments using global reference summaries. Specifically, we utilize global summarization queries and reference summaries to serve as a training corpus under the supervised setting. As there is only one global summarization query for each long document, we replicate it multiple times to match the quantity of self-supervised

training data, thus eliminating the impact of the quantity difference. We present the results on the WCEP and BookSum datasets in Figure 5, and the *Entropy* denotes the entropy of the similarity distribution between queries and node embeddings.

From Figure 5, it is evident that in the self-supervised setting, the loss is consistently lower than in the supervised setting. This suggests that the global reference summaries are highly correlated with many nodes, causing most nodes to exhibit a high semantic similarity with the global query. As a result, this confuses the model's optimization direction. Additionally, the entropy curve shows that the entropy in the supervised setting is consistently higher than in the self-supervised setting, indicating that the model struggles to select the most similar node. In contrast, the self-supervised label, derived from a specific part of a long document, contains more focused content and can more effectively guide the model's optimization direction.

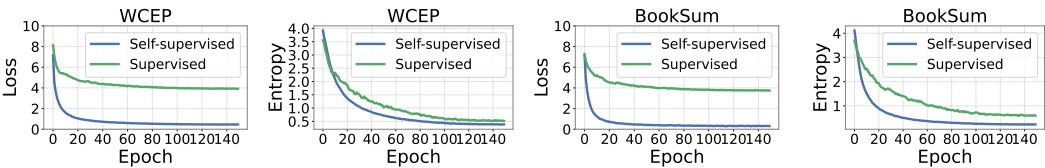

Figure 5: **Differences between self-supervised and supervised training w.r.t. loss and entropy**. We show the loss and entropy curve during training on the WCEP and BookSum datasets from left to right.

## 4 RELATED WORK

**Long-context Summarization using LLMs.** In recent years, LLMs have demonstrated their excellent capabilities in long-context modeling (Achiam et al., 2023; AI@Meta, 2024; Team et al., 2024; Jiang et al., 2024). Current approaches for summarizing long documents using LLMs are mainly divided into two categories: retrieval-augmented generation (RAG) (Ram et al., 2023; Yu et al., 2023) and long-context LLMs (GLM et al., 2024; Li* et al., 2023; Tworkowski et al., 2023). Long-context LLMs feature a large context window length to accommodate long-context inputs. However, they may suffer from severe performance degradation when accessing some local key details in the middle of long contexts (Liu et al., 2024b). In contrast, RAG emerges as a promising approach for cost-effective long-context summarization. By equipping with a retriever (Karpukhin et al., 2020; Robertson et al., 2009), RAG can first perform a relevance search based on user queries and then feed the retrieved text into LLMs for summary. For a recent example, GraphRAG (Edge et al., 2024) conducts query-focused summarization by setting up a graph index and detecting graph communities for summary generation.

Nevertheless, most current RAG approaches still focus on enhancing LLMs' reasoning and question-answering capabilities, which only require retrieving locally relevant information (Trivedi et al., 2022; Jiang et al., 2023b; Asai et al., 2023; Li et al., 2023; Zheng et al., 2023). In comparison, our proposed method stands out from these methods by focusing on LLMs' global summarization capability of long-context inputs.

**Historical Response Utilization of LLMs.** Little work has been done on this under-explored topic. Thought-retriever (Feng et al., 2024) saves the historical responses of each user-LLM interaction as high-level and informative thoughts to expand the retrieval corpus for future user queries. However, the rich correlation among thoughts is neglected, leaving room for further improvement.

Another line of work is the Chain-of-Thought (CoT), which is similar to our approach in terms of utilizing LLM historical responses and has been regarded as an effective means to enhance the reasoning ability of LLM during inference time in recent years. Few-shot CoT (Wei et al., 2022) and zero-shot CoT (Kojima et al., 2022) elicit intermediate reasoning paths by prompting LLMs with several demonstrations or just appending *"Let's think step by step."* Self-consistency (Wang et al., 2022b) samples diverse reasoning paths and conducts a majority vote to obtain the final answer. ToT (Yao et al., 2024) and GoT (Besta et al., 2024) take a further step by integrating a

tree or graph structure to manage its historical reasoning paths, enabling more flexible reasoning and reflection of LLMs. By memorizing the solution paradigms of various queries into different templates, BoT (Yang et al., 2024) pushes LLM's reasoning ability to a new level. Although the above-mentioned CoT-series approaches improve the reasoning capabilities of LLMs by utilizing chains of intermediate reasoning responses, they usually concentrate on one specific QA-type query and cannot generalize to and benefit other queries. Moreover, only a small number of historical reasoning responses are retained for the final generation, while most of the rest are just discarded.

**Graph-assisted Retrieval-augmented Language Models.** As one of the effective structures for modeling data relations, graphs have recently been used to enhance the performance of retrieval-augmented language models on various QA tasks. EtD (Liu et al., 2024a) features a graph neural network (GNN) to traverse a knowledge graph hop by hop to discover more relevant knowledge, thus enhancing LLM generation quality. GNN-RAG (Mavromatis & Karypis, 2024) learns to reason over graphs using GNNs, and retrieves answer candidates for a given question. PG-RAG (Liang et al., 2024) constructs pseudo-graphs with a retrieval indexer by prompting LLMs to organize document knowledge in a self-learning manner. G-RAG (Dong et al., 2024) proposes to rerank documents by learning graph representation on abstract meaning representation graphs, while GNN-Ret (Li et al., 2024b) refines semantic distances between documents and queries by modeling relationships among related passages. ToG (Sun et al., 2023) and KGP (Wang et al., 2024) treat LLMs as agents to traverse and reason over knowledge graphs in an iterative way, while RoG (Luo et al., 2023) first generates plans for retrieval and then conducts reasoning. G-Retriever (He et al., 2024) transforms retrieved knowledge subgraphs into graph embeddings by training a graph encoder and textualizes subgraphs to serve as inputs of LLMs.

Different from the above, GraphRAG (Edge et al., 2024) sets up a graph index for query-focused summarization, which is more relevant to our proposed approach. Compared with GraphRAG, which is time-consuming and suffers from huge computational costs, our proposed GoR is lightweight and only draws on a few LLM historical responses with efficient training to achieve competitive performance.

## 5 CONCLUSION

In this work, we introduce a method named *graph of records* (GoR) to improve long-context global summarization in retrieval-augmented generation (RAG) by utilizing LLM-generated historical responses. Intuitively, we establish connections between text chunks retrieved from long documents and LLM-generated historical responses to create a graph of records. To uncover complex correlations between these connections, we use a graph neural network and develop a BERTScore-based objective for self-supervised training. This allows for seamless supervision signal backpropagation between self-supervised reference summaries and node embeddings. In our experiments, we assess our proposed method on four long-context summarization datasets, and the results consistently demonstrate that GoR outperforms numerous baselines by a significant margin, highlighting its effectiveness.

Despite the superiority of our proposed method, GoR has some limitations. (1) Due to a limited budget, we only simulate and generate a small number of user queries, which may cause a bottleneck in further model optimization. (2) The simulated queries may not accurately reflect the real-world distribution, as they do not account for the possibility of users asking many meaningless questions. Therefore, a filtering process may be necessary, which we leave for future work.

ETHICS STATEMENT

The security of Large Language Models (LLMs) has always been a concern. Unfortunately, current LLMs sometimes produce harmful and biased information unexpectedly. Our proposed method uses LLMs to generate simulated queries and summary responses, which are only used to construct a graph of records and connect text chunks from long documents. However, more work is needed in real-world applications to ensure that LLMs' responses are reliable and harmless, so that they do not harm users.

REPRODUCIBILITY STATEMENT

In the experimental setup phase 3.1, we clearly describe the open-source LLMs used in this paper (Mixtral-8x7B-Instruct-v0.1 (Jiang et al., 2024) and LLaMA-2-7b-chat (Touvron et al., 2023)), the data preprocessing details (including RAG-related chunk size settings and dataset descriptions, etc.), and the training hyper-parameters. Moreover, we provide all the prompts used in this work in Appendix C. More detailed information can be found in the Appendix. The above description ensures the reproducibility of this work.

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

## A  EXPERIMENTAL DETAILS

### A.1  DATASET

We present dataset statistics in Table 3. Due to the limited budget, we randomly select training and test samples for the training and test set, and calculate the average input and output token lengths using the LLaMA-2 tokenizer (Touvron et al., 2023).

We evaluate our proposed method on four long-context summarization datasets, *i.e.*, AcademicEval (Feng et al., 2024), QMSum (Zhong et al., 2021), WCEP (Gholipour Ghalandari et al., 2020), and BookSum (Kryściński et al., 2021).

- **QMSum** (Zhong et al., 2021). QMSum is a query-based summarization dataset, which features lengthy meeting transcripts, specific queries, and general queries. Specific queries focus on query-based summarization, and general queries are questions that summarize the entire meeting transcript, such as "Summarize the whole meeting." We **only use** *"general queries"* for evaluating global summarization.

- **AcademicEval** (Feng et al., 2024). AcademicEval collects scientific papers from arXiv for abstract and related work writing. We use the abstract writing subset, which provides the main body of a paper as input and generates the predicted abstract.

- **WCEP** (Gholipour Ghalandari et al., 2020). WCEP is a multi-document summarization dataset about news events, which requires comprehensive consideration of the contents of multiple documents.

- **BookSum** (Kryściński et al., 2021). BookSum features long-form narrative summarization, which covers source documents from the literature domain and includes highly abstractive human-written summaries.

Table 3: **Dataset statistics**

| Dataset | #Train | #Test | Average Input Token Length | Average Output Token Length |
|---|---|---|---|---|
| QMSum (Zhong et al., 2021) | 162 | 30 | 17K | 0.1K |
| AcademicEval (Feng et al., 2024) | 400 | 30 | 13K | 0.3K |
| WCEP (Gholipour Ghalandari et al., 2020) | 400 | 30 | 11K | 0.05K |
| BookSum (Kryściński et al., 2021) | 400 | 30 | 16K | 1K |

## A.2 BASELINES

We present detailed descriptions of adopted baselines.

- **Node2Vec** (Grover & Leskovec, 2016). Node2Vec generates node embeddings for graphs by simulating biased random walks to capture both local and global structural properties of nodes.

- **BM25** (Robertson et al., 2009), **TF-IDF** (Ramos et al., 2003). BM25 ranks documents based on term frequency, inverse document frequency, and document length normalization, while TF-IDF evaluates the importance of a term in a document relative to a corpus by combining term frequency and inverse document frequency. In our experiments, the retrieval corpus is only the document chunks.

- **Contriever** (Izacard et al., 2021), **DPR** (Karpukhin et al., 2020), **Dragon** (Lin et al., 2023), **SBERT** (Reimers & Gurevych, 2019). Contriever is a self-supervised dense retriever that learns unsupervised document embeddings for information retrieval, DPR (Dense Passage Retriever) is a bi-encoder model that retrieves relevant passages by training on question-passage pairs, Dragon is a dense retrieval model optimized through diverse augmentation for generalizable dense retrieval, and SBERT (Sentence-BERT) is a modification of BERT that generates semantically meaningful sentence embeddings for tasks like similarity and clustering using a siamese network structure. In our experiments, the retrieval corpus is only the document chunks.

- **BM25+DPR**. BM25+DPR with Reciprocal Rerank Fusion is a hybrid retrieval method that combines the strengths of BM25's lexical matching and DPR's dense embeddings by reranking results from both models using a reciprocal rank fusion strategy to improve retrieval accuracy. In our experiments, the retrieval corpus is only the document chunks.

- **Gemma-8K** (Team et al., 2024), **Mistral-8K** (Jiang et al., 2023a). Gemma-8K and Mistral-8K are LLMs with relatively long context window lengths.

- **Full Context**. We feed all inputs to LLMs for summary generation. If the input length exceeds the context window limit, we randomly sample continuous text spans of maximum length multiple times to feed into LLMs and calculate the average result.
- **Thought-R** (Feng et al., 2024). Thought Retriever (Thought-R) generates thoughts for a series of simulated queries and appends them to the retrieval corpus as high-level knowledge.

## A.3 Additional Explanation on Training Objective

Given a graph that consists of document chunks and response nodes, we expect that the learned node embeddings $\mathbf{h}_v^{(L)}$ can adaptively reflect the semantic similarity to a given query $\mathbf{q}$. In other words, we expect that we can select the node $\mathbf{v}$ with the largest semantic similarity to $\mathbf{q}$ according to the formula $\mathrm{Sim}(\mathbf{q}, \mathbf{v}) = \mathbf{E_q}(\mathbf{q})^T \cdot \mathbf{h}_v^{(L)}$. To this end, we need to find out which node has the highest semantic similarity with $\mathbf{q}$ and use this as a supervision signal for model optimization. Therefore, we utilize BERTScore (Zhang et al., 2019) to obtain a node ranking list $\mathbf{M}_i$, which exactly serves as supervision signals.

$$\mathbf{M}_i = [\mathbf{h}_+^{(L)}, \mathbf{h}_1^{(L)}, \cdots, \mathbf{h}_{|\mathbf{C}|+|\mathbf{T}|}^{(L)}] \tag{8}$$

For contrastive loss in Equation 5, we regard $\mathbf{h}_+^{(L)}$ as the positive and $[\mathbf{h}_1^{(L)}, \cdots, \mathbf{h}_{|\mathbf{C}|+|\mathbf{T}|}^{(L)}]$ as the negatives to conduct contrastive learning (van den Oord et al., 2018). For a given query $\mathbf{q}$, the contrastive learning objective will bring $\mathbf{E_q}(\mathbf{q})$ and $\mathbf{h}_+^{(L)}$ closer in the semantic embedding space while increasing the distance between $\mathbf{E_q}(\mathbf{q})$ and $[\mathbf{h}_1^{(L)}, \cdots, \mathbf{h}_{|\mathbf{C}|+|\mathbf{T}|}^{(L)}]$ (the symbols are simplified here for convenience of description).

Similarly, in Equation 6, we expect to impose a stricter constraint on the learned node embedding based on the node ranking list $\mathbf{M}_i$. Although Equation 5 shortens the semantic distance between $\mathbf{E_q}(\mathbf{q})$ and $\mathbf{h}_+^{(L)}$, it does not take into account the relative ranking between negative samples. For example, the semantic similarity between $\mathbf{E_q}(\mathbf{q})$ and $\mathbf{h}_1^{(L)}$ is higher than that between $\mathbf{h}_2^{(L)}$. Formally, given $\mathbf{h}_i^{(L)}, \mathbf{h}_j^{(L)} \in \mathbf{M}_i$ that satisfies $\mathrm{rank}(\mathbf{h}_j^{(L)}) > \mathrm{rank}(\mathbf{h}_i^{(L)})$, Equation 6 will explicitly optimize in the direction of $\mathbf{E_q}(\mathbf{q})^\top \mathbf{h}_j^{(L)} < \mathbf{E_q}(\mathbf{q})^\top \mathbf{h}_i^{(L)}$, thus imposing stricter constraints to the pair-wise ranking.

## A.4 Additional Implementation Details

In the stage of graph construction, due to the number and randomness of the simulated queries, there may be some isolated nodes, and we just keep them in the graph with self-loop edges. During model optimization, BERTScore is pre-computed for efficient training. We present hyper-parameters on QMSum, AcademicEval, WCEP, and BookSum datasets in Table 4.

For metrics, we adopt Rouge-1 (R-1), Rouge-2 (R-2), and Rouge-L (R-L) (Lin, 2004) to assess the text alignment between the reference summaries and the predicted content generated by our proposed method. If a global summarization query has multiple reference summaries, we calculate the Rouge-L/1/2 of the predicted summary and all references, respectively, and take the maximum value as the final evaluation result. We follow this setting in all experiments, including the baseline evaluation.

## A.5 Additional Experimental Results on Inference Efficiency

To investigate the inference efficiency of our proposed method, we conduct extensive experiments on the WCEP (Gholipour Ghalandari et al., 2020) dataset and present the results w.r.t. the inference time per query in Table 5.

Table 4: **Hyper-parameters**

| Datasets | QMSum | AcademicEval | WCEP | BookSum |
|---|---|---|---|---|
| #GAT Layers | 2 | 2 | 2 | 2 |
| #GAT Heads | 4 | 4 | 4 | 4 |
| Batch Size | 32 | 32 | 32 | 32 |
| Epoch | 150 | 150 | 150 | 150 |
| Learning Rate | 1e-3 | 1e-3 | 1e-3 | 1e-3 |
| Hidden Dimension | 768 | 768 | 768 | 768 |
| Dropout Rate | 0.2 | 0.0 | 0.1 | 0.2 |
| Loss Coefficient $\alpha$ | 0.9 | 0.6 | 0.7 | 0.2 |

From Table 5, we can draw the following conclusion. (1) Since GoR's only trainable module is GNN, GoR's inference efficiency is very high, and almost no additional noticeable latency is introduced. (2) Although GoR's inference time is longer than some baselines, it only increases by a few hundred milliseconds. Considering the significant performance improvement brought by GoR in Table 1, this tiny time overhead is almost negligible in practical applications.

Table 5: **Inference Efficiency Analysis**. Since the LLM used in our experiment is consistent, we ignore the inference time brought by the LLM itself.

| Baselines | Node2Vec | BM25 | Contriever | SBERT | BM25+DPR | Thought-R | GoR (ours) |
|---|---|---|---|---|---|---|---|
| Inference Time (s) | 9.4 | 0.02 | 0.20 | 0.01 | 0.04 | 0.3 | 0.58 |

## B  CASE STUDY

In this section, we provide a case study of GoR and other baseline methods' summarization.

---

**The Summary Generated by GoR**

An asteroid called 1999 KW4 will make a close approach to Earth this weekend, with its own small moon in tow. The asteroid is estimated to be around 1.5 km wide and will be roughly 3.2 million miles from Earth at its closest point. NASA astronomers are interested in studying the asteroid due to its close proximity and the fact that it is a binary system, meaning it has its own moon. The last time the asteroid will make a close approach to Earth will be in 2036. Additionally, a recent study using data from NASA's Kepler Space Telescope suggests that comets may have delivered water to Earth, as the ratio of two types of water molecules on comets matches that in Earth's oceans. The new algorithm used in the study is more sensitive to small planets the size of Earth and could help in the search for Earth-like planets.

---

**The Summary Generated by Contriever**

Asteroid 2019 JH7 recently flew past Earth, and NASA observed that the asteroid's trajectory falls under the "Earth Close Approach" category. The observations made by NASA's Jet Propulsion Laboratory (JPL) in Pasadena, California, suggest that comets could have delivered water to Earth. The study found that the ratio of two types of water molecules on the comet matches that in the Earth's oceans, which could imply that comets delivered a large fraction of water to the outer reaches of the solar system. The observations made using data from the SOFIA telescope suggest that all comets could have a heavy-to-regular water ratio similar to Earth's oceans, which could imply that comets delivered some water to Earth. Previously, measuring this ratio was difficult, and ground and space telescopes could only study this level of detail in comets when they pass near Earth.

**Reference Summary**

Binary Aten asteroid (66391) 1999 KW4 and its minor-planet moon make their closest-ever recorded flyby of Earth at 3.2 million miles away. The asteroid will approach even closer at 0.0155 AU (2,320,000 km) from Earth in 2036, and is the largest asteroid to approach Earth until (4953) 1990 MU in June 2027.

From the above example, we can draw conclusions. (1) GoR summarizes several keywords that appear in the reference summary, such as "1999 KW4" and "3.2 million miles", etc., but Contriever fails to extract this crucial information. (2) From a global perspective, the summary generated by GoR is more relevant and consistent with the reference summary. However, the summary generated by Contriever focuses too much on local details and ignores the main idea of the original article.

## C   LLM Prompts

In this section, we present LLM prompts used in GoR, including user query simulation and RAG prompts.

### C.1   LLM Prompts for User Query Simulation

**Prompt for User Query Simulation**

You are a great questioner of any text, and are adept at asking valuable and insightful questions. Your goal is to generate 1 summary question for the text provided below. The generated summary question should try to simulate the tone of human questions as much as possible, and make sure that the generated question must be interrogative sentences and a summary question. Important! Please make sure this text must be a complete and non-redundant answer to the generated summary question. Please directly output the generated summary question, do not output irrelevant text.

DOCUMENT:
{document}

## C.2 LLM PROMPTS FOR RAG

> **RAG Prompt**
>
> Refer to the following supporting materials and answer the question with brief but complete explanations.
>
> SUPPORTING MATERIALS:
> {materials}
>
> QUESTION:
> {question}

## D BROADER IMPACTS

In the era of LLMs, countless interactions take place between users and these models on a daily basis, resulting in the generation of a vast amount of historical responses. Our proposed method demonstrates that these historical responses hold significant potential and can be effectively leveraged to further improve the quality of future responses generated by LLMs. By analyzing and reusing these past outputs, we can not only refine and enhance the overall performance of the models but also reduce computational overhead. This approach highlights the untapped value of historical data in optimizing response generation while making the process more efficient and resource-friendly.

