# OpenReview forum: "Graph of Records: Boosting Retrieval Augmented Generation for Long-context Summarization with Graphs"
_ICLR.cc/2025/Conference — Submitted to ICLR 2025_

### Official Review · Reviewer_mVtZ · 2024-10-28

**Soundness:** 2
**Presentation:** 3
**Contribution:** 3
**Rating:** 6
**Confidence:** 4

**Summary:**

This paper presents GoR, a retrieval-augmented generation method aimed at enhancing the summarization of long documents. The authors propose a novel approach to distilling the knowledge of large language models (LLMs) to improve retrieval by incorporating both historical LLM responses and text chunks into a structured graph. Their second key contribution is a comprehensive framework for training a graph neural network (GNN)-based retriever on this graph, learning to rank chunk and response nodes using contrastive and ranking objectives. Experimental comparisons with traditional retrievers and long-context LLMs demonstrate the advantages of GoR in long-document summarization.

**Strengths:**

1. Using LLM's parametric knowledge—referred to as historical responses in this paper—to enhance retrieval-augmented generation (RAG) is a valuable addition to the static corpus used in traditional RAG. The authors claim to be the first to apply this approach to long-document summarization.
2. The experimental results are promising, though some analyses may not fully address the research question.
3. The paper is well-written and easy to follow.

**Weaknesses:**

1. The analyses lack sufficient convincing evidence. See the “Questions” section for more details.
2. The graph construction process is complex, which could potentially increase inference time—a critical factor in RAG systems.

**Questions:**

1. The authors discuss the impact of different GNN architectures; however, a more fundamental research question remains: does the constructed structure itself actually improve performance? A follow-up question is: does the LLM's historical responses are really helpful, as claimed by the authors? I understand these are challenging to address, but the paper could benefit from empirical studies, such as ablation experiments on the constructed graph (e.g., ablating the response nodes).

2. A small issue for the presentation. In the analysis on L424, a more informative metric on the x-axis might be the number of tokens used for training. Increasing the number of simulated queries also scales the number of responses generated by the LLM, meaning the model could benefit from both the queries and the additional response tokens.

3. I would appreciate a comparison of inference time per query for the baselines in the main table. This information is valuable for readers interested in inference efficiency for deployment.

4. Regarding "Supervised Training on Global Summarization Queries," are you referring to a baseline where LLaMA-2-7B-chat is fine-tuned using supervised training on global summarization queries and reference summaries?

5. Would it make sense to add query nodes between the response and context nodes in the graph? My intuition is that query nodes could help the model better understand logical constraints between context and response (e.g., negation).

---

> ### Author Response · Authors · 2024-11-17
> **Response to Reviewer mVtZ**
>
> Thanks for your valuable feedback and appreciation.
>
> > **Question 1**
>
> Regarding “Does the constructed structure itself actually improve performance?” and “Does the LLM's historical responses are really helpful?” mentioned by the reviewer, **we have conducted experiments about the impact or the benefit of the graph structure and response nodes**. In **Table 1 (Page 7)**, the results of “Contriever” and “GoR” (**Lines 335 and 345**) **indicate that the integration of response nodes brings significant improvement to the summarization performance**. The experiment of "Contriever" does not introduce LLM-generated responses as response nodes; that is, its retrieval corpus is only the document itself without a graph structure. Since the default retriever of GoR is contriever, it can be seen from this experimental result that LLM-generated responses are helpful. Moreover, in **Table 2 (Page 7)**, we further **demonstrate that the model optimization on the constructed graph improves the ROUGE by a significant margin** compared with the “w/o train” (**Lines 353 and 358**).
>
> Overall, **in Line 335 in Table 1 (Page 7)**, the retrieval corpus is only the document itself; **in Line 353 in Table 2 (Page 7)**, the retrieval corpus is the document and the response nodes. By comparing the results of these two experiments, we can demonstrate that **(1) the integration of response nodes brings significant improvement to the summarization performance; (2) the model optimization on the constructed graph further improves the ROUGE by a significant margin**.
>
> We will describe the experimental settings of this part in more detail in the revised version.
>
> > **Question 2**
>
> Thanks for your suggestions, and we will modify the description in the revised version.
>
> > **Question 3**
>
> Thanks for mentioning this point, and we would also like to make some clarifications about the inference efficiency and graph construction process.
>
> As for the graph construction process, we just follow RAG's retrieve-then-generate paradigm, that is, connect the retrieved chunks with the LLM-generated responses, **which is intuitive, simple, and natural**.
>
> As for inference efficiency, **we provide evaluation results on inference time per query in the following table** (since the LLM used in our experiment is consistent, we ignore the inference time brought by the LLM itself).
>
> |Baseline|Node2Vec|BM25|Contriever|SBERT|BM25+DPR|Thought-R|GoR (ours)|
> |:-:|:-:|:-:|:-:|:-:|:-:|:-:|:-:|
> Inference Time (s)|9.4|0.02|0.20|0.01|0.04|0.3|0.58|
>
> Since GoR's only trainable module is GNN, GoR's inference efficiency is very high, and **almost no additional noticeable latency is introduced**. Although GoR's inference time is longer than some baselines, **it only increases by a few hundred milliseconds**. Considering the significant performance improvement brought by GoR, **this tiny time overhead is almost negligible in practical applications**. We will add these results in the revised version.
>
> > **Question 4**
>
> No, **the only trainable module of GoR is GNN. We didn’t fine-tune LLMs in our work.**
>
> Since GoR utilizes simulated queries to conduct self-supervised optimization, we would like to compare it with using global summarization queries (e.g., “Summarize the contents of this report.”) for supervised training to illustrate the rationality of GoR's training pipeline design. The reference summaries of global summarization queries are highly correlated with many nodes, which causes most nodes to exhibit a high semantic similarity (measured by BERTScore) with the global query, confusing the model’s optimization direction. In contrast, the self-supervised label, derived from a specific part of a long document, contains more focused content and can more effectively guide the model’s optimization direction (**Line 436-442**).
>
> > **Question 5**
>
> Thanks for mentioning this interesting point. Actually, in our early exploration, we have also considered adding query nodes, regarding query, document chunk, and response as three different node types, and using heterogeneous graph neural networks for representation learning. Although this design seems reasonable, it makes the graph larger and more difficult to optimize, and the performance is almost the same as the current design of GoR. Therefore, we adopt the current, more concise design method, which is both simple and effective.
>
> Thank you again for your constructive feedback, and we look forward to further engaging in discussions to improve the quality and impact of our work.

---

> > ### Comment · Reviewer_mVtZ · 2024-11-24
> >
> > Thanks for your time and effort in the rebuttal. I think your response addresses most of my confusion and concern about this paper. Therefore, I decide to revise my score accordingly.

---

> ### Author Response · Authors · 2024-11-25
> **Response to Reviewer mVtZ**
>
> Thank you for your thoughtful and constructive feedback. We are pleased to hear that our responses have addressed most of your concerns. We are committed to incorporating the suggested changes in our revisions to further enhance the manuscript.

---

### Official Review · Reviewer_RomL · 2024-10-30

**Soundness:** 2
**Presentation:** 2
**Contribution:** 3
**Rating:** 5
**Confidence:** 3

**Summary:**

This paper proposes a novel method called GoR (Graph of Records) to enhance RAG in long-context global summarization tasks. GoR utilizes historical responses generated by LLMs to construct a graph structure that connects retrieved text chunks with their corresponding responses. This paper further employs a graph neural network and BERTScore-based self-supervised training to reveal correlations between them. Experimental results show that GoR outperforms 12 baselines across four long-context summarization datasets, demonstrating its effectiveness in improving summarization quality.

**Strengths:**

Innovative Use of Historical Responses: The paper introduces a novel approach by leveraging LLM-generated historical responses for enhancing RAG, which is largely neglected by existing methods. This approach enriches the summarization process, potentially increasing the relevance and depth of generated summaries.

**Weaknesses:**

__I.__ Computational Efficiency Evaluation: The paper lacks experimental validation of computational efficiency. The reliance on LLM-generated responses and retrieved chunk graphs, combined with the incorporation of graph neural networks and BERTScore-based objectives, could introduce substantial computational overhead.

__II.__ Dependency on Data Quality: The effectiveness of GoR may rely heavily on the quality and coherence of the historical responses generated by LLMs. Inconsistencies in these responses could impact the model's overall summarization performance.

__III.__ The quality of the presentation is below ICLR 2025 standards. For example:

  a. The format of the references should be consistent to ensure neatness and professionalism. For instance, the names of conferences should be uniformly presented either in abbreviations or full names, rather than a mixture of both.

  b. For the line 265, it should be written as “we only use “general queries” for evaluating”, and the symbol use is wrong.

**Questions:**

Please see weakness

---

> ### Author Response · Authors · 2024-11-17
> **Response to Reviewer RomL**
>
> Thanks for your valuable feedback.
>
> > **Computational Efficiency Evaluation**
>
> We fully understand the reviewer’s concerns about computational efficiency, and we would like to make clarifications by providing evaluation results on inference time per query in the following table (since the LLM used in our experiment is consistent, we ignore the inference time brought by the LLM itself).
>
> |Baseline|Node2Vec|BM25|Contriever|SBERT|BM25+DPR|Thought-R|GoR (ours)|
> |:-:|:-:|:-:|:-:|:-:|:-:|:-:|:-:|
> Inference Time (s)|9.4|0.02|0.20|0.01|0.04|0.3|0.58|
>
> Since GoR's only trainable module is GNN, GoR's inference efficiency is very high, and **almost no additional noticeable latency is introduced**. Although GoR's inference time is longer than some baselines, **it only increases by a few hundred milliseconds**. Considering the significant performance improvement brought by GoR, **this tiny time overhead is almost negligible in practical applications**. We will add these results in the revised version.
>
> Additionally, LLM-generated responses and retrieved chunks are naturally what the RAG system needs, which does not bring any additional computational overhead to GoR. Moreover, since GoR's trainable module is only GNN, the training cost is also very low, which only takes about half an hour to complete the training (~400(graphs)*30(simulated queries)=12000 training samples) on a single RTX3080 and achieve good results.
>
> > **Dependency on Data Quality**
>
> Thanks for mentioning this point, and we would like to make some clarifications.
>
> Low-quality responses are almost impossible to completely avoid. The responses generated by LLMs inevitably contain some incoherent or ambiguous text due to LLM hallucinations or other reasons[1,2].
>
> **Overall, according to Tables 1 and 2 (Page 7), GoR does improve the performance by a significant margin** compared to several baselines on global long-context summarization tasks, demonstrating the effectiveness of our method.
>
> **Moreover, the design of GoR can naturally alleviate this issue effectively**. Perhaps some of these responses are low-quality texts, but thanks to the construction of the graph and the use of BERTScore, the semantic similarity between these noise texts (or nodes) and the golden answer of a given training query will be low, so they will be ranked later in the node ranking list, which will not affect the direction of the entire model optimization (**Line 210-255**). **Overall, the utilization of BERTScore during the training phase enables GoR to be immune to most of the noisy texts**.
>
> > **Presentation Quality**
>
> Thanks for mentioning these typos. For the mixing of conference abbreviations and full names in Reference and improper use of double quotes (if this is what you mean, we are a little confused), we will fix these typos in the revised version.
>
>
> [1] A Survey on Hallucination in Large Language Models: Principles, Taxonomy, Challenges, and Open Questions. 2023
>
> [2] Siren's Song in the AI Ocean: A Survey on Hallucination in Large Language Models. 2023
>
>
> Thank you again for your constructive feedback, and we look forward to further engaging in discussions to improve the quality and impact of our work.

---

> ### Author Response · Authors · 2024-11-25
> **Response to Reviewer RomL**
>
> Dear Reviewer RomL,
>
> We recognize that the timing of this discussion period may not align perfectly with your schedule, yet we would greatly value the opportunity to continue our dialogue before the deadline approaches.
>
> We hope that our responses and additional experiments have effectively addressed your concerns. We truly appreciate all the valuable advice we have received, and we are pleased to share that other reviewers have kindly recognized our improvements by raising their ratings or confidence. This acknowledgment reflects the positive impact of our collaborative efforts in enhancing the quality of the paper.
>
> Could you let us know if your concerns have been adequately addressed? If you find that your concerns have been resolved, we would appreciate it if you could reconsider the review score.
>
> Thanks!

---

> > ### Comment · Reviewer_RomL · 2024-11-26
> > **Response to authors**
> >
> > Thank you for addressing my concerns. I will adjust my score accordingly to reflect the improvements.

---

> > > ### Author Response · Authors · 2024-11-26
> > > **Response to Reviewer RomL**
> > >
> > > Dear reviewer RomL,
> > >
> > > Thank you for your prompt response. Could you let us know if you have any other concerns? If so, could you kindly specify the remaining concerns? We will try our best to solve them in the next few days. And we kindly request you consider further increasing the score.
> > >
> > > Thank you.

---

> > > > ### Author Response · Authors · 2024-11-30
> > > > **Response to Reviewer RomL**
> > > >
> > > > Dear reviewer RomL,
> > > >
> > > > We would like to express our sincere appreciation for your positive opinions and constructive review of our paper on the occasion of Thanksgiving. We apologize for intruding during your busy schedule, but as the discussion period is near its end, we would like to ensure our response aligns with your expectations and addresses your concerns. If you find that your concerns have been resolved, we would appreciate it if you could reconsider the review score.
> > > >
> > > > Wishing you a joyful Thanksgiving,
> > > >
> > > > Best regards,
> > > >
> > > > Authors

---

### Official Review · Reviewer_XmHJ · 2024-11-03

**Soundness:** 2
**Presentation:** 1
**Contribution:** 2
**Rating:** 6
**Confidence:** 4

**Summary:**

This paper presents a novel retrieval-augmented generation (RAG) approach for long-context document summarization. The method leverages a graph neural network (GNN) to retrieve relevant information from structured textual nodes. These nodes comprise two types of content: (1) original text chunks from the source document and (2) intermediate summaries previously generated by large language models (LLMs). To optimize the GNN retriever, the authors propose aligning the GNN's representations with semantic similarities from textual encoder, creating a more semantically-aware retrieval mechanism.

Key contributions include:

- A hierarchical text representation using GNNs that combines both source text and intermediate summaries
- A novel training objective that aligns GNN embeddings with semantic similarity metrics
- An end-to-end framework that integrates retrieval and generation for long-document summarization

**Strengths:**

1. Introduces an innovative approach to integrate LLMs' intermediate summaries with original document chunks in a structured graph representation
2. Implements an efficient sparse connectivity strategy using top-K similar textual chunks, reducing computational complexity while maintaining information flow
3. This hierarchical structure effectively bridges local and global document understanding
4. Develops a well-motivated semantic alignment mechanism for the GNN by leveraging BERTScore-based similarities
5. Employs a dual-objective training strategy combining contrastive learning and pairwise ranking loss
6. This approach ensures the GNN's representations capture meaningful semantic relationships beyond surface-level textual similarity
7. Provides a thorough analysis of existing long-context summarization approaches, clearly identifying their limitations
8. Effectively positions the work within two key research streams: retrieval-augmented generation and graph-based document processing
9. Demonstrates clear technological advancement over previous methods through thoughtful experimental design

**Weaknesses:**

1. Clarity and Presentation Issues:
- **Problem Definition** : Lacks a dedicated subsection that formally defines long-context summarization; Missing explicit comparison between GoR and existing RAG approaches for long-context summarization in the introduction; Would benefit from a clear positioning diagram or framework overview

- **Technical Clarity** : Graph construction description (lines 147-194) lacks sufficient detail and clear visualization; Equation 3's retrieval mechanism is ambiguous about whether it includes LLM-generated responses or only original document chunks; The relationship between the proposed contrastive learning approach and BERTScore needs clearer explanation; The semantic alignment objective would benefit from step-by-step derivation


2. Limited Evaluation Metric: Over-reliance on ROUGE metrics for evaluation; Absence of crucial evaluation metrics: Human evaluation for factual consistency and coherence; BERTScore for semantic similarity assessment; Coverage metrics for long-document comprehension.

3. Insufficient Analysis of Critical Components: Lacks detailed analysis of edge creation strategy between document chunks and LLM responses; Insufficient investigation of the impact of LLM-generated responses; No comparative analysis showing the benefit of including historical LLM responses.

**Questions:**

1. What are Ec and Eq context encoders in line 224-230? Are they also optimized by eq. 7?
2. The paper mentioned BERTscore many times in this paper but how does BERTscore contribute to the GoR?
3. In both contrastive learning and ranking loss, what is the motivation of  choosing the positive samples based on the semantic similarity from the encoder?
4. Provide a case study of GoR and other baseline methods' summarization.

---

> ### Author Response · Authors · 2024-11-17
> **Response to Reviewer XmHJ (1/n)**
>
> Thanks for your valuable feedback.
>
> > **Question 1: What are Ec and Eq context encoders in line 224-230? Are they also optimized by eq. 7?**
>
> $E_c$ and $E_q$ are the context encoder and query encoder from the retriever, which are responsible for encoding document context and user questions, respectively. **We have described it in Section “2.1 PRELIMINARIES” (Line 114-116)**. $E_c$ and $E_q$ will not be optimized by $Eq.7$. In GoR, the only module that can be optimized (or be trained) is GNN (**Line 254-256**).
>
> > **Question 2: The paper mentioned BERTscore many times in this paper but how does BERTscore contribute to the GoR?**
>
> **We thank the reviewer for expressing his understanding and appreciation of the proposed BERTScore-based objective in Points 4, 5, and 6 of “Strengths”**. In GoR, given a user query $q$, our ultimate goal is to make the learned node embeddings adaptively reflect the similarity with the query embedding $E_q(q) $ by taking the complicated correlations among nodes into account. However, there is a backpropagation gap between the golden answer (i.e., label or reference) of the query $q$ and the text contained in nodes (since we need to provide the model with supervision signals, i.e., given the golden answer of the query $q$, which node is most semantically relevant to it?) (**Line 199-209**). Therefore, **BERTScore is employed to bridge this gap, which quantifies the semantic similarity between the golden answer and the text contained in nodes**. In this way, given the golden answer of a query $q$, we can rank all nodes based on BERTScore, and the node ranking list is further utilized in the calculation of the loss function (**Line 211-255**). For example, the first-ranked node is used as a positive sample for contrastive learning, forcing the node embedding to be closer to the query embedding in the semantic embedding space (measured by the dot product of embeddings, **Line 116**) after model optimization.
>
> > **Question 3: In both contrastive learning and ranking loss, what is the motivation of choosing the positive samples based on the semantic similarity from the encoder?**
>
> **We thank the reviewer for expressing his understanding and appreciation of GoR in Points 4, 5, and 6 of “Strengths”**. We **don’t** choose the positive sample based on the semantic similarity from the encoder $E_c$ or $E_q$. Given a training query and its golden answer, we first calculate BERTScore between the golden answer and the text contained in nodes. Then, we rank the nodes based on the calculated BERTScore and choose the one with the largest value as the positive (**Line 211-238**). The larger the BERTScore value, the closer the node is to the golden answer semantically, which is aligned with our optimization goal.

---

> ### Author Response · Authors · 2024-11-17
> **Response to Reviewer XmHJ (2/n)**
>
> > **Question 4: Provide a case study of GoR and other baseline methods' summarization.**
>
> Due to the word limit, we provide a case study of GoR and Contriever's summarization.
>
> **GoR**
>
> An asteroid called 1999 KW4 will make a close approach to Earth this weekend, with its own small moon in tow. The asteroid is estimated to be around 1.5 km wide and will be roughly 3.2 million miles from Earth at its closest point. NASA astronomers are interested in studying the asteroid due to its close proximity and the fact that it is a binary system, meaning it has its own moon. The last time the asteroid will make a close approach to Earth will be in 2036. Additionally, a recent study using data from NASA's Kepler Space Telescope suggests that comets may have delivered water to Earth, as the ratio of two types of water molecules on comets matches that in Earth's oceans. The new algorithm used in the study is more sensitive to small planets the size of Earth and could help in the search for Earth-like planets.
>
> **Contriever**
>
> Asteroid 2019 JH7 recently flew past Earth, and NASA observed that the asteroid's trajectory falls under the "Earth Close Approach" category. The observations made by NASA's Jet Propulsion Laboratory (JPL) in Pasadena, California, suggest that comets could have delivered water to Earth. The study found that the ratio of two types of water molecules on the comet matches that in the Earth's oceans, which could imply that comets delivered a large fraction of water to the outer reaches of the solar system. The observations made using data from the SOFIA telescope suggest that all comets could have a heavy-to-regular water ratio similar to Earth's oceans, which could imply that comets delivered some water to Earth. Previously, measuring this ratio was difficult, and ground and space telescopes could only study this level of detail in comets when they pass near Earth.
>
> **Reference Summary**
>
> Binary Aten asteroid (66391) 1999 KW4 and its minor-planet moon make their closest-ever recorded flyby of Earth at 3.2 million miles away. The asteroid will approach even closer at 0.0155 AU (2,320,000 km) from Earth in 2036, and is the largest asteroid to approach Earth until (4953) 1990 MU in June 2027.
>
> **From the above example, we can draw conclusions.**
>
> (1) GoR summarizes several keywords that appear in the reference summary, such as "1999 KW4" and "3.2 million miles", etc., but Contriever fails to extract this crucial information.
>
> (2) From a global perspective, the summary generated by GoR is more relevant and consistent with the reference summary. However, the summary generated by Contriever focuses too much on local details and ignores the main idea of the original article.

---

> ### Author Response · Authors · 2024-11-17
> **Response to Reviewer XmHJ (3/n)**
>
> > **Problem Definition**
>
> **We thank the reviewer for appreciating that our work provides a detailed comparison with related work and positions our work well (Points 7 and 8 of “Strengths”)**. We will reiterate some of the problem definitions of our work below.
>
> >> **Long-context Summarization Definition**
>
> We follow most related works[1,2,3,4,6] to describe long-context summarization in the “Related Work” section, avoiding the introduction of too many mathematical symbols while clarifying the difference between GoR and other work, **as the reviewer mentioned in Points 7 and 8 of “Strengths”**.
>
> >> **Comparison between GoR and existing RAG approaches for long-context summarization in the introduction**
>
> To make the paper well-structured, we have described it in detail in the “Related Work” section (**Line 457-474, 492-510**). We also thank the reviewer for appreciating our clear and thorough description of the differences between GoR and other related works (**Points 7 and 8 of “Strengths” mentioned by the reviewer**).
>
> > **Technical Clarity**
>
> **We thank the reviewer for appreciating that our work provides clear technological advancement over previous methods through thoughtful experimental design (Points 1-6 and 9 of “Strengths”)**. We will reiterate some of the technical details of our work below.
>
> >> **Graph construction description and Equation 3's retrieval mechanism**
>
> **As the reviewer mentioned in Points 1, 2, and 3 of “Strengths,” we thank the reviewer for the positive feedback and appreciation of our clear description of the graph construction (Line 147-194)**. We will re-describe and emphasize some key details as follows.
>
> The graph construction is simple and intuitive. Inspired by the retrieve-then-generate paradigm of RAG, we just connect the retrieved text chunks and the LLM-generated response (**Line 147-194**). For instance, suppose text chunks $c_1$ and $c_2$ are retrieved, which are fed into LLMs to obtain the response $r$. We create an edge between $c_1$ and $r$, $c_2$ and $r$, respectively. Through performing RAG on the simulated queries, we can finally obtain a graph for each long document. Moreover, **we have provided a detailed visualization of graph construction in Figure 2**, which describes the intricate correlations between document chunks and response nodes.
>
> As for Equation 3, the retrieval corpus is dynamic and includes LLM-generated responses. **We have described it clearly in our paper (Line 159-161), and Figure 2 also shows that the retrieval corpus includes LLM-generated responses (e.g., $r_2$ and $c_m$ => $r_6$)**.
>
> >> **The relationship between contrastive learning and BERTScore**
>
> **As the reviewer mentioned in Points 4 and 5 of “Strengths,” we thank the reviewer for the positive feedback and appreciation of our model design on contrastive learning and BERTScore (Line 210-255)**. We have answered this question in the previous Q2 and Q3. Please refer to them.
>
> > **Limited Evaluation Metric**
>
> Thanks for your constructive advice.
>
> **Considering that GoR has utilized BERTScore in the model optimization process, we only use ROUGE in the evaluation stage for a fair comparison.** ROUGE is a commonly adopted metric for evaluating automatic text summarization and can be also used to measure coverage and density of information, which is widely adopted by a wide range of works[1,2,3,4,5,6] (**these works have also relied exclusively on ROUGE as the main metric for summarization tasks**)
>
> **In line with related studies [2,3,4,6], we followed established practices and conducted automatic evaluations without incorporating human evaluations**. We acknowledge the value of human evaluation for assessing factors like factual consistency and coherence, and we plan to incorporate this in future work to provide more comprehensive insights.
>
> **Regarding coverage metrics, these are less suitable for long-context summarization tasks, particularly for open-ended queries such as "Summarize the contents of this report."** Unlike tasks with specific text chunk labels, long-context summarization involves considering the entire document as a potential source of relevant information. In such cases, all document chunks may contribute to the summary, making coverage metrics less effective for evaluation.
>
> We appreciate your suggestions and will consider extending our evaluation methodology, including the addition of human evaluation, in future iterations of this work.
>
>
> [1] Efficient Attentions for Long Document Summarization. NAACL. 2021.
>
> [2] Long-Span Summarization via Local Attention and Content Selection. ACL. 2021.
>
> [3] A Discourse-Aware Attention Model for Abstractive Summarization of Long Documents. NAACL. 2018.
>
> [4] Big Bird: Transformers for Longer Sequences. NeurIPS. 2020.
>
> [5] GSum: A General Framework for Guided Neural Abstractive Summarization. NAACL. 2021.
>
> [6] DYLE: Dynamic Latent Extraction for Abstractive Long-Input Summarization. ACL. 2022.

---

> ### Author Response · Authors · 2024-11-17
> **Response to Reviewer XmHJ (4/n)**
>
> > **Lack detailed analysis of edge creation strategy between document chunks and LLM responses**
>
> **We would like to thank the reviewer for appreciating our model and experiment design in Points 2 and 9 in “Strengths”**. We follow the “retrieve-then-generate” paradigm of RAG to create edges between document chunks and LLM responses (**Line 157-194**). Suppose we have retrieved three document chunks $c_1$, $c_2$, and $c_3$, which are then fed into LLMs to obtain the response $r$。$r$ is generated from $c_1$, $c_2$, and $c_3$, so we create an edge between $c_1$ and $r$, $c_2$ and $r$, $c_3$ and $r$. **The motivation for edge creation is intuitive and natural and has been proven to be effective in our experiments** (**Tables 1 and 2, Page 7**). While our current edge creation strategy has proven successful, we acknowledge the potential for further refinement and plan to explore more advanced strategies in future work.
>
> > **Insufficient investigation of the impact of LLM-generated responses; No comparative analysis showing the benefit of including historical LLM responses.**
>
> **We would like to thank the reviewer for appreciating our thoughtful experiment design in Point 9 in “Strengths”**. We have conducted comprehensive experiments about the impact or benefit of LLM-generated responses. In **Table 1 (Page 7)**, the results of “Contriever” and “GoR” (**Line 335 and 345**) **indicate that LLM-generated responses bring significant improvement to the summarization performance**. The experiment of "Contriever" does not introduce LLM-generated responses; that is, its retrieval corpus is only the document itself. Since the default retriever of GoR is contriever, it can be seen from this experimental result that LLM-generated responses are effective. Moreover, in **Table 2 (Page 7)**, we further **demonstrate that the model optimization on the constructed graph improves the ROUGE by a significant margin** compared with the “w/o train” (**Lines 353 and 358**).
>
> Overall, **in Line 335 in Table 1 (Page 7)**, the retrieval corpus is only the document itself; **in Line 353 in Table 2 (Page 7)**, the retrieval corpus is the document and the response nodes. By comparing the results of these two experiments, we can demonstrate that **(1) the integration of response nodes brings significant improvement to the summarization performance; (2) the model optimization on the constructed graph further improves the ROUGE by a significant margin.**
>
> We will describe the experimental settings of this part in more detail in the revised version.
>
> Thank you again for your constructive feedback, and we look forward to further engaging in discussions to improve the quality and impact of our work.

---

> ### Author Response · Authors · 2024-11-25
> **Response to Reviewer XmHJ**
>
> Thank you for your thoughtful and constructive feedback. We are pleased to hear that our responses have addressed most of your concerns. We are committed to incorporating the suggested changes in our revisions to further enhance the manuscript.

---

### Official Review · Reviewer_ESSc · 2024-11-04

**Soundness:** 3
**Presentation:** 3
**Contribution:** 3
**Rating:** 6
**Confidence:** 5

**Summary:**

The paper introduces the Graph of Records (GoR), a novel method designed to enhance retrieval-augmented generation (RAG) systems for long-context summarization using graph structures. RAG systems improve factuality and relevance in summarization by retrieving relevant content chunks and generating summaries based on those chunks. GoR innovates on traditional RAG by constructing a graph of linked nodes where historical user queries, LLM-generated responses, and retrieved text chunks are interconnected. This graph structure is then refined using a graph neural network (GNN), enabling the model to capture complex relationships between text chunks and responses. By employing a BERTScore-based self-supervised learning objective, GoR aligns node embeddings with semantic relevance to improve summarization accuracy. The model outperforms several baselines across four long-context summarization datasets (e.g., QMSum, AcademicEval), showcasing its effectiveness in creating more comprehensive summaries.

**Strengths:**

1. The paper is well-structured, with clear explanations and logically organized sections, making it easy for readers to follow the methodology and findings.

2. The authors test GoR on four long-context summarization datasets, and the comprehensive evaluation against a variety of baselines (including both sparse and dense retrievers) demonstrates the robustness and generalizability of the method.

3. The proposed approach builds upon existing RAG techniques, making it relatively easy to implement and reproduce, which is valuable for the research community and practical applications.

**Weaknesses:**

I didn’t find significant weaknesses in this paper. The entire architecture is built on existing modules, making the proposed framework both sound and replicable. However, this reliance on established methods might also be a limitation, as the framework feels less innovative or exciting despite being well-presented with informative experimental results.

**Questions:**

I have no further questions about this paper. I am inclined to give it a weak positive score and will review the feedback from other reviewers before finalizing my assessment.

---

> ### Author Response · Authors · 2024-11-17
> **Response to Reviewer ESSc**
>
> Thanks for your valuable feedback and appreciation of the soundness, presentation, and contribution to our work. We appreciate your insights and would like to further address the concerns you raised about the innovative aspects of our framework.
>
> While it is true that the GoR incorporates existing modules, **the novelty of our work lies in the design of the algorithm pipeline, which addresses a critical and under-explored challenge in the utilization of LLM historical responses** (**Line 42-45**).
>
> Specifically, our approach unlocks the potential of leveraging historical responses generated by LLMs in retrieval-augmented generation (RAG) systems—an aspect that has not been systematically explored to date.
>
> In the context of the increasing reliance on LLMs, our framework capitalizes on the vast repository of historical interactions between users and models. By analyzing and reusing these past outputs, we demonstrate that these historical responses are not only valuable but can significantly improve the quality of future responses. This innovation not only enhances LLM performance but also introduces a resource-efficient methodology, reducing computational overhead without compromising response quality.
>
> Moreover, the use of established modules contributes to the replicability and robustness of our framework, enabling researchers and practitioners to easily adapt and extend our methods to related tasks. The simplicity of the architecture lowers the barrier for adoption and facilitates broader impact in real-world applications.
>
> Our work paves the way for future research by highlighting the untapped potential of historical data in response optimization. This contribution is not limited to our current submission but sets a foundation for exploring resource-aware, data-driven strategies in the field.
>
> Thank you again for your constructive feedback, and we look forward to further engaging in discussions to improve the quality and impact of our work.

---

### Author Response · Authors · 2024-11-17
**Summary of the major revision**

We thank the reviewers for the thorough and detailed reviews on our submission. We summarize major changes that we have made below. All changes are marked in blue in the updated submission.

- We fixed typos mentioned by reviewer RomL and mVtZ

- We provided a case study of GoR and other baseline methods' summarization in Appendix B (Page 17), mentioned by reviewer XmHJ

- We added additional experimental results on inference efficiency in Appendix A.5 and Table 5 (Page 16 and 17), mentioned by reviewer RomL and mVtZ

- We provided a further explanation of the training objective of GoR in Appendix A.3 (Page 16), mentioned by reviewer XmHJ

- We added a detailed baseline description in Appendix A.2 (Page 15)

- We added a "Broader Impacts" section in Appendix D (Page 19)


We respectfully request the reviewers consider these improvements based on the constructive feedback provided. We believe these enhancements strengthen the paper and provide valuable insights to the community, and we hope for consideration of increasing the scores of our submission. Thank you for your thoughtful consideration.

---

> ### Author Response · Authors · 2024-11-23
> **Thanks for your feedback**
>
> Dear reviewers,
>
> We would like to thank you again for taking the time to review our paper and provide valuable comments. We understand that you are busy and that the review process may take some time. We look forward to your response and further engaging in discussions to improve the quality and impact of our work.
>
> Thanks.

---

### Meta-Review · Area_Chair_EF2P · 2024-12-20

**Metareview:**

**Summary:**
The paper proposes Graph of Records (GoR), a method to enhance RAG for long-context summarization by leveraging LLM-generated historical responses, which are often neglected. GoR constructs a graph that links retrieved text chunks with their corresponding LLM responses, enabling the discovery of complex correlations through a GNN. A self-supervised training framework using a BERTScore-based objective ranks node embeddings by their relevance to global summaries, supported by contrastive and pairwise ranking losses. Evaluated on several long-context summarization datasets, the proposed method outperforms baselines, demonstrating its effectiveness in modeling and utilizing historical response data to improve summarization quality.

**Strength:**
- The framework is sound, replicable, and built on established modules, ensuring reliability. The experimental results are informative and well-presented.
- Hierarchical structure seems to effectively bridge local and global document understanding

**Weakness:*
- The reliance on established methods might also be a limitation.
- Missing a formal definition of long-context summarization and a clear comparison with existing RAG approaches.
- Heavy reliance on ROUGE metrics, with no human evaluation or additional metrics like BERTScore or coverage for long-document comprehension.

**Additional Comments On Reviewer Discussion:**

There was active discussion between the authors and reviewers during the rebuttal process. While I believe most of the concerns were adequately addressed, I think the evaluation relies too heavily on ROUGE scores. In text summarization, it is well-known that ROUGE scores do not necessarily indicate high-quality summaries, as they fail to account for issues such as hallucination, omission, and overall coherence [1,2]. Since ROUGE primarily measures lexical overlap, I find it difficult to fully trust the significance of the results presented in the paper, given that no additional evaluation metrics were utilized. At least one of the recent automated metric should be included. Therefore I am leaning to reject this paper.

Here is reference papers saying that Rouge is not aligned with human judgement:

[1] Evaluating the Factual Consistency of Abstractive Text Summarization, EMNLP 2020

[2] G-Eval: NLG Evaluation using GPT-4 with Better Human Alignment, EMNLP 2024

[3] Fine-grained Summarization Evaluation using LLMs, ACL 2024

---

### Decision · Program_Chairs · 2025-01-22

Reject